# Optimal Neural Program Synthesis from Multimodal Specifications

## Abstract

Multimodal program synthesis, which leverages different types of user input to synthesize a desired program, is an attractive way to scale program synthesis to challenging settings; however, it requires integrating noisy signals from the user (like natural language) with hard constraints on the program's behavior. This paper proposes an *optimal neural synthesis* approach where the goal is to find a program that satisfies user-provided constraints while also maximizing the program's score with respect to a neural model. Specifically, we focus on multimodal synthesis tasks in which the user intent is expressed using combination of natural language (NL) and input-output examples. At the core of our method is a top-down recurrent neural model that places distributions over abstract syntax trees conditioned on the NL input. This model not only allows for efficient search over the space of syntactically valid programs, but it allows us to leverage *automated program analysis* techniques for pruning the search space based on infeasibility of partial programs with respect to the user's constraints. The experimental results on a multimodal synthesis dataset (STRUCTUREDREGEX) show that our method substantially outperforms prior state-of-the-art techniques in terms of accuracy and explores fewer states during search.

## 1 Introduction

In recent years, there has been a revolution in machine learning-based *program synthesis* techniques for automatically generating programs from high-level expressions of user intent, such as input-output examples (Balog et al., 2017; Chen et al., 2019a; Devlin et al., 2017; Ellis et al., 2019; Kalyan et al., 2018; Shin et al., 2018) and natural language (Yaghmazadeh et al., 2017; Dong & Lapata, 2016; Rabinovich et al., 2017; Yin & Neubig, 2017; Desai et al., 2016; Wang et al., 2018). Many of these techniques use deep neural networks to consume specifications and then perform model-guided search to find a program that satisfies the user. However, because the user's specification can be inherently ambiguous (Devlin et al., 2017; Yin et al., 2018), a recent thread of work on *multimodal synthesis* attempts to combine different types of cues, such as natural language and examples, to allow program synthesis to effectively scale to more complex problems. Critically, this setting introduces a new challenge: how do we efficiently synthesize programs with a combination of hard and soft constraints from distinct sources?

In this paper, we formulate multimodal synthesis as an *optimal synthesis* task and propose an optimal synthesis algorithm to solve it. The goal of optimal synthesis is to generate a program that satisfies any hard constraints provided by the user while also maximizing the score under a learned neural network model that captures noisy information, like that from natural language. In practice, there are *many* programs that satisfy the hard constraints, so this maximization is crucial to finding the program that actually meets the user's expectations: if our neural model is well-calibrated, a program that maximizes the score under the neural model is more likely to be the user's intended program.

Our optimal neural synthesis algorithm takes as input multimodal user guidance. In our setting, we train a neural model to take natural language input that can be used to guide the search for a program consistent with some user-provided examples. Because our search procedure enumerates programs according to their score, the first enumerated program satisfying the examples is guaranteed to be optimal according to the model. A central feature of our approach is the use of a tree-structured neural model, namely the *abstract syntax network (ASN)* (Rabinovich et al., 2017), for constructing

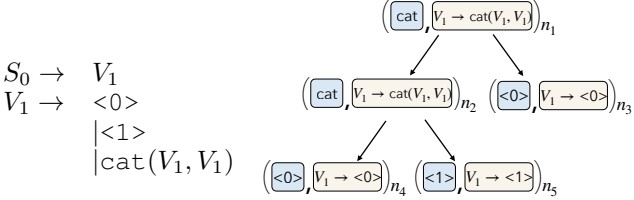

$$S_0 \rightarrow \quad V_1$$
$$V_1 \rightarrow \quad \texttt{<0>}$$
$$\quad \quad |\texttt{<1>}$$
$$\quad \quad |\texttt{cat}(V_1, V_1)$$

Figure 1: Example grammar for a simple language.

Figure 2: Example of an AST derivation of `cat(cat(<0>,<1>),<0>)`. Blue boxes represent symbols and yellow boxes represent productions.

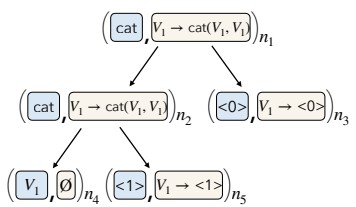

Figure 3: Example of a partial program. $n_4$ is a leaf node with non-terminal symbol $V_1$.

syntactically valid programs in a top-down manner. The structure of the ASN model restricts search to programs that are syntactically correct, thereby avoiding the need to deal with program syntax errors (Kulal et al., 2019), and it allows us to search over programs in a flexible way, without constraining a left-to-right generation order like seq2seq models do. More importantly, the use of top-down search allows us to more effectively leverage *automated program analysis* techniques for proving infeasibility of partial ASTs. As a result, our synthesizer can prune the search space more aggressively than prior work and significantly speed up search. While our network structure and pruning techique are adapted from prior work, we combine them and generalize them to this optimal neural synthesis setting in a new way, and we show that our general approach leads to substantial improvements over previous synthesis approaches.

We implement our method in a synthesizer called OPSYNTH and evaluate it on the challenging STRUCTUREDREGEX dataset (Ye et al., 2020a) for synthesizing regular expressions from linguistically diverse natural language descriptions and positive/negative examples. We compare our approach against a range of approaches from prior work and ablations of our own method. OPSYNTH achieves substantial gain over past work by solving 59.8% (resp. 46.9%) of the programs of Test (resp. Test-E) set in STRUCTUREDREGEX by exploring on average 560 (resp. 810) states, which surpasses previous state-of-the-art by 11.7% (resp. 10.9%) with $23\times$ (resp. $18\times$) fewer states.

## 2  PROBLEM FORMULATION

**Context-free grammar.**   In this work, we assume that the syntax of the target programming language $L$ is specified as a context-free grammar $\mathcal{G} = (V, \Sigma, R, S_0)$ where $V$ is a set of non-terminals, $\Sigma$ is the set of terminal symbols, $R$ is a set of productions, and $S_0$ is the start symbol. We use the notation $s$ to denote any *symbol* in $V \cup \Sigma$. The grammar in Figure 1 has two nonterminals ($S_0$ and $V_1$) and three terminals (cat, <0>, and <1>). To simplify presentation in the rest of the paper, we assume that each grammar production is of the form $v \rightarrow f(s_0, \ldots, s_n)$ where $f$ is a language construct (e.g., a constant like 0 or a built-in function/operator like cat, +, etc.).

We represent programs in terms of their abstract syntax trees (AST). We assume that every node $n$ in an abstract syntax tree is labeled with a grammar symbol $s$ (denoted $\mathcal{S}(n)$), and that every node is labeled with a production $r \in R$ (denoted $\mathcal{R}(n)$) that indicates which CFG production was used to assign a terminal symbol for that node (if applicable). Figure 2 shows an AST representation of the program `cat(cat(<0>,<1>),<0>)` generated using the simple grammar shown in Figure 1.

**Partial programs.**   For the purposes of this paper, a *partial program* is an AST in which some of the nodes are labeled with non-terminal symbols in the grammar (see Figure 3). For a *complete program*, all node labels are terminal symbols. We use the notation EXPAND$(P, l, r)$ to denote replacing leaf $l$ with production $r$, which adds $n$ nodes $s_1, \ldots, s_n$ to the tree corresponding to the yield of $r$.

**Consistency with examples.**   In this paper, we focus on multimodal synthesis problems where the user provides a logical specification $\phi$ in addition to a natural language description. More concretely, we focus on logical specifications that are in the form of positive and negative examples on the program behavior. Each example is a pair $(x, y)$ such that, for a positive example, we have $P(x) = y$ for the target program $P$, and for a negative example, we have $P(x) \neq y$. Given a set of examples

$\mathcal{E} = \mathcal{E}^+ \cup \mathcal{E}^-$ and program $P$, we write $P \models \mathcal{E}$, if we have $P(x) = y$ for every positive example in $\mathcal{E}^+$ and we have $P(x) \neq y$ for every negative example in $\mathcal{E}^-$. If $P$ is a partial program, $P \not\models \phi$ indicates that there is no completion of $P$ that satisfies the specification $\phi$.

**Optimal multimodal synthesis problem.** A second input to our multimodal synthesis problem is a natural language description of the task. We define a model $M_\theta(P \mid N)$ that yields the probability of a given program conditioned on the natural language description (described in Section 3). Given a programming language $L$ specified by its context-free grammar, a logical specification $\phi$ (e.g., a set of positive and negative examples), natural language description $N$ and a model $M_\theta$, our goal is to find the most likely program in the language satisfying the constraints:

$$\underset{P \in L \ \wedge \ P \models \phi}{\arg\max} \ M_\theta(P \mid N) \tag{1}$$

## 3 OPTIMAL NEURAL SYNTHESIS ALGORITHM

We consider a class of models $M_\theta$ that admit efficient optimal synthesis. Any model with the properties described in this section can be plugged into our synthesis algorithm (Section 3.2).

**Definition 3.1. (AST Path)** Given a node $n$ in a partial program $P$, we define the AST path $\pi(P, n) = ((n_1, i_1), \ldots, (n_k, i_k))$ to be a sequence of pairs $(n_j, i_j)$ where (1) AST node $n_{j+1}$ is the $i_j$'th child of AST node $n_j$ and (2) the $i_k$'th child of $n_k$ is $n$. For instance, for the partial program in Figure 3, we have $\pi(P, n_4) = ((n_1, 1), (n_2, 1))$.

**Definition 3.2. (Concrete/Inconcrete nodes)** Given a partial program $P$, we define the concrete nodes of $P$ as $\mathcal{C}(P)$ to be the nodes which have production rules assigned to them. The inconcrete nodes $\mathcal{I}(P)$ are the non-terminal leaf nodes whose production rules haven't been determined and need to be fill in in order to form a complete program.

Given a partial program $P$, we define the probability of generating $P$ as the product of the probabilities of applying the productions labeling each node in the AST. There are a number of possible ways we could factor and parameterize this distribution, including PCFGs, where the distribution depends only on the parent, or as sequence models over a pre-order traversal of the tree (Dong & Lapata, 2016; Yin & Neubig, 2017; Polosukhin & Skidanov, 2018). We choose the following factorization, similar to that used in Abstract Syntax Networks (ASN) (Rabinovich et al., 2017), where a production rule depends on the derivation path leading to that nonterminal:

$$p_\theta(P \mid N) = \prod_{n \in \mathcal{C}(P)} p_\theta(\mathcal{R}(n) \mid \pi(P, n), N) \tag{2}$$

The chief advantage of this factorization is that the score of a partial program is **invariant to the derivation order of that program**: two derivations of the same tree $P$ that differ only in the order that branches were generated are still assigned the same probability, allowing for flexibility in the search process. Second, for a partial program $P$, the distribution over rules of every unexpanded non-terminal leaf node does not depend on the others', which allows the estimation of the *upper bound* (maximum possible probability) of the complete programs that can be derived from $P$. Specifically, we define the upper bound of the complete programs that can possibly be derived from a partial program $P$ as:

$$u_\theta(P \mid N) = p_\theta(P \mid N) \cdot \prod_{n \in \mathcal{I}(P)} \max_r p_\theta(r \mid \pi(P, n), N). \tag{3}$$

This bound incorporates the known probabilities of concrete nodes as well as the minimum cost of filling inconcrete non-terminals, and thus more accurately estimates the cost of the optimal complete program given this partial program. A sequence model traversing the tree with a fixed order cannot estimate such an upper bound as the probabilities of inconcrete nodes are not known.

### 3.1 NEURAL MODEL

We instantiate the neural model defined above using a simplified version of ASN (Rabinovich et al., 2017), which respects the $p_\theta(\mathcal{R}(n) \mid \pi(P, n), N)$ factorization for the production of each node in the tree. Figure 4 illustrates how ASN recursively computes the probability of labeling a node $n$ as $\mathcal{R}(n)$.

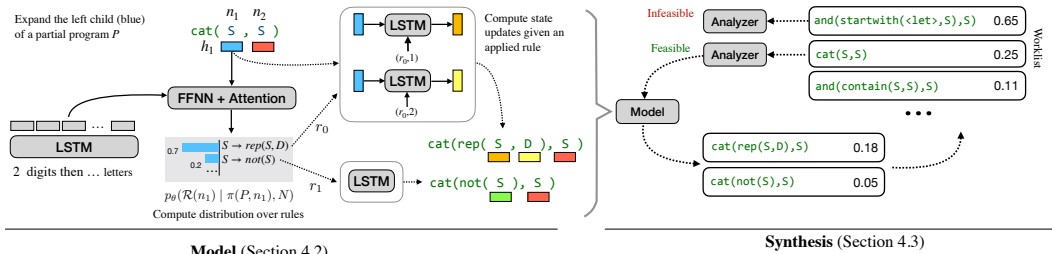

Figure 4: Left: our neural model. A vector $h_i$ associated with a nonterminal is used to predict a distribution over grammar rules. Each rule instantiates new nonterminals which receive updated vectors based on LSTMs. Right: partial programs are taken from the worklist, analyzed, and expanded, then the new partial programs are added to the worklist.

---

**Algorithm 1** Synthesis Algorithm

---

1: **procedure** OPSYNTH($\mathcal{G}, \phi, N, M_\theta$)
    **input:** A CFG $\mathcal{G} = (V, \Sigma, R, S_0)$, specification $\phi$, natural language $N$ and model $M_\theta$
    **output:** Complete program $P$ with highest probability under $M_\theta$ that satisfies $\phi$, or $\perp$
2:      $\mathcal{Q} := \{(S_0, 1)\}$;
3:      **while** $\mathcal{Q} \neq \emptyset$ **do**
4:         $(P, \rho) := \mathcal{Q}.\text{dequeue}()$;             ▷ upper bound $\rho$ associated with the partial program $P$
5:         **if** $\texttt{Infeasible}(P, \phi)$ **then** $\texttt{continue}$;
6:         **if** $\texttt{IsConcrete}(P)$ **then return** $P$;
7:         $l := \texttt{SelectLeaf}(P)$
8:         **for** $r \in \texttt{Supp}(M_\theta(\pi(P, l), N))$ **do**
9:            $P' := \texttt{Expand}(P, l, r)$
10:           $\mathcal{Q}.\text{add}((P', u_\theta(P'|N))$
11:     **return** $\perp$;

---

Consider the partial program $\texttt{cat}(\mathcal{S}(n_1), \mathcal{S}(n_2))$; we need to define the probability distribution over legal productions on the first node $n_1$: $p_\theta(\mathcal{R}(n_1) \mid \pi(P, n), N) = p_\theta(\mathcal{R}(n_1) \mid \{(\texttt{cat}, 1)\}, N)$.

We encode the AST path using an LSTM (Hochreiter & Schmidhuber, 1997). Define $\text{LSTM}(h_0, (r_j, i_j))$ to be an LSTM with initial state $h_0$ and which, at each timestep, consumes a tuple consisting of a node $n_j$ and a parent-child index $i_j$ (i.e., an element in $\pi(P, n)$).[1] We embed each tuple $(n_j, i_j)$ by $W_{\mathcal{R}(n_j), i_j}$, where $W$ is specialized to the rule and position. Then: $h_{\text{root}} = \text{LSTM}(N)$ and $h_n = \text{LSTM}(h_{\text{root}}, \pi(P, n))$ where $\text{LSTM}(N)$ denotes an encoding of the natural language. The hidden state $h_n$ encodes both the user's NL specification as well as where we are in the parse tree, allowing us to model which grammar symbol should be likely at this position.

Given this hidden state $h_n$, the probability for each production rule at node $n$ is computed using a feedforward neural network (FFNN) module and attention over the NL input:

$$p_\theta(\cdot \mid \pi(P, n), N) = \text{softmax}(\text{FFNN}(h_n; \text{Attn}(h_n, \text{LSTM}(N))))$$

During search, each $\texttt{Expand}$ operation instantiates a node $n$ with each possible rule according to the probabilities above, then computes the hidden states for any new nonterminals using the LSTM.

## 3.2 SYNTHESIS

In this section, we describe a search algorithm to solve the optimal neural synthesis problem defined in Equation 1. The key idea is to maintain a priority list $\mathcal{Q}$ of partial programs, ranked according to the upper bound ($u_\theta(P)$) probability of the complete programs that can be derived from this partial program. Then, in each iteration of the search procedure, we pick the highest upper bound partial

---

[1]This abstraction allows our LSTM to implement the hidden state computation of the "constructor" module from Rabinovich et al. (2017). Our production rule model follows the "primitive" and "composite type" modules.

$$\frac{\mathsf{Root}(P) = n \quad \mathcal{S}(n) \in V}{P \hookrightarrow (y = \top, y = \bot)} \text{ (a)}$$

$$\frac{\mathsf{Root}(P) = n \quad n_i \in \mathsf{Children}(P) \quad \mathsf{Subtree}(P, n_i) \hookrightarrow (\psi_i^+(y, \mathbf{x}), \psi_i^-(y, \mathbf{x}))}{P \hookrightarrow (\exists \mathbf{z}.(\Phi^+(\mathcal{S}(n))) \wedge \bigwedge_i \psi_i^+[z_i/y]), \exists \mathbf{z}.(\Phi^-(\mathcal{S}(n))) \wedge \bigwedge_i \psi_i^-[z_i/y])} \text{ (b)}$$

$$\frac{P \hookrightarrow (\psi^+(y, \mathbf{x}), \psi^-(y, \mathbf{x})) \quad \mathbf{UNSAT}(\bigwedge_{(\mathbf{i},o) \in \mathcal{E}^+} \psi^+[o/y, \mathbf{i}/\mathbf{x}] \wedge \bigwedge_{(\mathbf{i},o) \in \mathcal{E}^-} \neg \psi^-[o/y, \mathbf{i}/\mathbf{x}])}{P \not\models (\mathcal{E}^+, \mathcal{E}^-)} \text{ (c)}$$

Figure 5: Inference rules describing procedure INFEASIBLE$(P, \phi)$ for specification $\phi$ consisting of positive examples $\mathcal{E}^+$ and negative examples $\mathcal{E}^-$. Rules (a)-(b) of the form $P \hookrightarrow (\phi^+, \phi^-)$ generate a pair of logical formulas over- and under- approximating the semantics of partial program $P$. The notation $\psi[z/y]$ denotes substituting variable $y$ with $z$ in formula $\psi$.

program $P$ in $\mathcal{Q}$, check its feasibility using program analysis, and if it is feasible, expand one of the non-terminals in $P$ using the applicable CFG productions. Since complete programs are dequeued from $\mathcal{Q}$ in decreasing order of their probability according to $M_\theta$, the first complete program that satisfies $\phi$ is guaranteed to be optimal under $M_\theta$ (Proof in the in Appendix); thus, our algorithm is guaranteed to return an optimal program if a solution exists.

**Infeasibility pruning**  Our top-down search allows us to exploit program analysis techniques to prune the search space, by determining whether $P$ is infeasible with respect to the user's hard constraints. A common way of doing this is to use well-known *abstract interpretation* techniques from the programming languages literature to approximate program semantics (Cousot & Cousot, 1977; Nielson et al., 2015). In particular, given a partial program $P$, the idea behind the feasibility checking procedure is to generate a pair of logical formulas $(\psi^+, \psi^-)$ over- and under-approximating $P$'s semantics respectively. If there is any positive example $e^+ \in \mathcal{E}^+$ that is inconsistent with $\psi^+$, then the partial program is infeasible. Similarly, if there is any negative example $e^- \in \mathcal{E}^-$ that satisfies $\psi^-$, we can again conclude that $P$ must be infeasible.

Figure 5 describes our feasibility checking procedure in terms of inference rules, where rules (a) and (b) generate a pair of over- and under-approximations of the program, and rule (c) checks feasibility of these approximations with respect to the provided examples. Here, free variables $\mathbf{x}$ in the formula represent program inputs, and free variable $y$ represents the program output. The existentially quantified variables $\mathbf{z}$ corresponds to values of sub-expressions. The first rule states that "holes" (i.e., non-terminals) in the partial program are over-approximated using $y = \top$ meaning the sub-program can return anything, and they are under-approximated using $y = \bot$, meaning that the sub-program returns nothing. The second rule is used to (recursively) construct an approximation of a sub-AST rooted at node $n$. This rule utilizes a pair of mappings $\Phi^+, \Phi^-$ where $\Phi^+$ (resp. $\Phi^-$) gives an over-approximating (resp. under-approximating) semantics for each language construct. In rule (b), each child formula $\psi_i^+, \psi_i^-$ must be satisfied as well as the parent formula, and these are unified by a shared set of new existentially-quantified variables.

The final rule uses the generated over- and under-approximations of the partial program to check feasibility. In particular, we conclude that the partial program is infeasible if there is any positive example $e^+ \in \mathcal{E}^+$ that is inconsistent with $\psi^+$ or any negative example $e^- \in \mathcal{E}^-$ that satisfies $\psi^-$.

## 4 EXPERIMENTAL SETUP

We evaluate our synthesizer on the STRUCTUREDREGEX dataset for multimodal synthesis of regular expressions. This dataset contains 3520 labeled examples, including an NL description, positive/negative examples, and the target regex. We choose this dataset for our evaluation because (1) it is only the dataset containing both examples and NL where the NL description is written by humans, and (2) this dataset is quite challenging, with existing techniques achieving under 50% accuracy.

**Implementation Details**  As stated in Section 3.1, our model is an Abstract Syntax Network tailored to fit the regex DSL used in STRUCTUREDREGEX. We train our neural model to maximize the log

likelihood of generating ground truth regexes given the NL using the Adam optimizer (Kingma & Ba, 2015), stopping when the performance on dev set converges. More details are in the Appendix.

We implement the infeasibility checking procedure for our regex DSL by encoding the semantics of each operator in the theory of strings (Liang et al., 2014). Since all existentially quantified variables in the resulting formula can be eliminated through substitution, the resulting constraints are of the form $s \in R$ (or $s \notin R$) where $s$ is a string constant and $R$ is a regular expression. Thus, we can check the satisfiability of these formulas using the Bricks library (Møller, 2017). The supplementary material describes both the semantics of the DSL constructs as well as an example showing how to generate the encoding for a given partial program.

Because of our infeasibility check, the order of expanding non-terminals can impact the efficiency of our search. We experimented with several methods of selecting a leaf node to expand, including pre-order traversal, choosing high-level nodes first, and choosing lowest-entropy nodes first. Pre-order traversal seemed to work best; more details about the expansion order can be found in Appendix.

**Baselines**   We compare our method against three programming-by-example (**PBE-only**) baselines, ALPHAREGEX (Lee et al., 2016), DEEPCODER (Balog et al., 2017), and ROBUSTFILL (Devlin et al., 2017). ALPHAREGEX is an enumerative regex synthesizer that uses breadth-first search to find regexes that are consistent with the examples. Both DEEPCODER and ROBUSTFILL are neural program synthesis approaches. DEEPCODER places distribution over constructs and terminals based on examples, and uses this distribution to carry out DFS search, whereas ROBUSTFILL uses beam search to autoregressively build programs.

We further compare our method against prior multimodal program synthesis techniques, SKETCH (Ye et al., 2020b) and TREESEARCH (Polosukhin & Skidanov, 2018) with appropriate tuning of the hyperparameters and the SKETCH synthesizer for this setting. We do not compare against SKETCHADAPT (Nye et al., 2019) because it relies on the assumption that every program consistent with examples is the gold program, which does not hold in our setting.

We also consider two NL-to-code models, Seq2Seq and TranX (Yin & Neubig, 2017), which we modify to filter out complete programs that are inconsistent with the examples (Chen et al., 2020; Polosukhin & Skidanov, 2018). A more sophisticated baseline (Ye et al., 2020a) uses example-guided pruning by filtering the beam at every timestep during search. We adopt these more sophisticated baselines proposed in  Ye et al. (2020a) to allow a fair comparison. Implementation details of all our baselines are in the Appendix.

We refer to our Optimal Synthesis approach as **OPSYNTH**. We also show ablations: $\text{ASN}^{+\mathcal{P}}$ (ASN with our pruning during beam search), and $\text{OPSYNTH}^{-\mathcal{P}}$ to further demonstrate the benefits of our approach over models like Polosukhin & Skidanov (2018) that do not use such pruning. Finally, we also consider an extension denoted as $\textbf{OPSYNTH}^{+\mathcal{R}}$, which extends OPSYNTH with the ATTENTION A MODEL from ROBUSTFILL (Devlin et al., 2017), which encodes the examples $\phi$ using another set of LSTM layers. To combine these signals, we define the probability of applying rule $r$ on $n$ as:

$$p_\theta(r|n, P, N) = \text{softmax}(\text{FFNN}(h_n; \text{Attn}(h_n, \text{context}(N)); \text{Attn}(h_n, \text{context}(\phi)))).$$

## 5   RESULTS AND ANALYSIS

In the following experiments, we evaluate our approach based on two criteria: (1) accuracy, defined as the fraction of solved synthesis tasks, and (2) efficiency, defined in terms of the number of partial programs searched. Note that we define efficiency in this way because the bottleneck is applying the EXPAND function on partial programs and symbolic evaluation of these partial programs rather than the neural net computation.

**Main Results**   Our main results are shown in Table 1. We report results on two test sets from STRUCTUREDREGEX; Test-E is annotated by a distinct set of annotators from the training set.

As shown in the top part of Table 1, pure PBE approaches (top) do poorly on this dataset due to not utilizing NL. These approaches either fail to find a regex consistent with the examples within a time limit of 90 seconds or the synthesized regex is semantically different from the target one. Thus, the comparison against PBE-only approaches demonstrates the importance of using a model that places distributions over programs conditioned on the NL description.

Table 1: Fraction of solved benchmarks (%Sol), fraction of benchmarks where we find a I/O-consistent program (%Cons), average number of states explored (#St), and average time used in seconds (Time).

| Approach | Test | | | | Test-E | | | |
|---|---|---|---|---|---|---|---|---|
| | %Sol | %Cons | #St | Time | %Sol | %Cons | #St | Time |
| AlphaRegex | 3.6 | 51.8 | $1.4_{10^6}$ | 51.0 | 3.5 | 49.6 | $1.4_{10^6}$ | 53.8 |
| DeepCoder | 1.1 | 6.2 | $7.4_{10^4}$ | 84.7 | 1.3 | 6.0 | $6.8_{10^4}$ | 86.2 |
| RobustFill | 3.5 | 39.4 | $1.9_{10^3}$ | 21.1 | 3.5 | 38.4 | $2.0_{10^3}$ | 22.1 |
| SKETCH | 45.2 | 75.4 | $3.1_{10^3}$ | 18.4 | 29.8 | 62.8 | $3.5_{10^3}$ | 21.5 |
| TREESEARCH | 48.7 | 69.8 | — | 13.2 | 31.1 | 56.1 | — | 19.1 |
| Seq2Seq$^{+\mathcal{P}}$ | 48.2 | 78.2 | $1.3_{10^4}$ | 66.5 | 36.0 | 64.3 | $1.5_{10^4}$ | 76.8 |
| TranX$^{+\mathcal{P}}$ | 53.1 | 87.8 | $5.6_{10^3}$ | 31.4 | 38.1 | 77.4 | $6.4_{10^3}$ | 36.1 |
| ASN$^{+\mathcal{P}}$ | 58.0 | 87.8 | $1.3_{10^3}$ | 13.6 | 45.8 | 78.2 | $1.4_{10^3}$ | 15.1 |
| OPSYNTH | **59.8** | 83.9 | $\mathbf{5.6_{10^2}}$ | 6.5 | **46.9** | 75.5 | $\mathbf{8.1_{10^2}}$ | 9.6 |
| OPSYNTH$^{-\mathcal{P}}$ | 55.3 | 74.7 | — | 8.4 | 43.1 | 62.7 | — | 11.5 |
| OPSYNTH$^{+\mathcal{R}}$ | 58.0 | 82.1 | $5.7_{10^2}$ | 6.8 | 44.1 | 74.8 | $8.2_{10^2}$ | 9.8 |

The second and third part of Table 1 shows results from prior multimodal neural synthesis approaches and NL-to-code models augmented with example-based pruning (Ye et al., 2020a). SKETCH slightly outperforms TREESEARCH, solving 45% and 30% of the Test and Test-E set respectively. Seq2Seq$^{+\mathcal{P}}$ and TranX$^{+\mathcal{P}}$, which perform beam search guided by the Seq2Seq and TranX models but also check feasibility of partial programs before adding them to the beam, outperform these other techniques: TranX$^{+\mathcal{P}}$ outperforms Seq2Seq$^{+\mathcal{P}}$ and solves 53% of the benchmarks on Test and 38% for Test-E.

The last part of Table 1 provides results about OPSYNTH and its ablations. OPSYNTH achieves a substantial improvement over TranX$^{+\mathcal{P}}$ and is able to solve approximately 60% of benchmarks in Test and 47% in Test-E. In addition to solving more benchmarks, OPSYNTH also explores only a fraction of the states explored by TranX$^{+\mathcal{P}}$, which translates into improving synthesis time by roughly an order of magnitude.

We now compare OPSYNTH against three of its ablations. OPSYNTH$^{-\mathcal{P}}$ does not use program analysis to prune infeasible partial programs (hence, we do not report explored states as a measure of runtime), and ASN$^{+\mathcal{P}}$ is similar to OPSYNTH except that it uses beam search (with beam size 20) combined with example-based pruning. Both the program analysis component and optimal search are important: without these, we observe both a reduction in accuracy and an increase in the number of states explored. The last row in Table 1 shows an extension of OPSYNTH described in Section 4 where we incorporate the ROBUSTFILL model. We find that ROBUSTFILL is ineffective on its own, and incorporating it into our base synthesizer actually decreases performance. While such neural-guided PBE approaches (DEEPCODER (Balog et al., 2017) and ROBUSTFILL (Devlin et al., 2017)) have been successful in prior work, they do not appear to be effective on this challenging task, or not necessary in the presence of strong natural language hints. Additionally, these models both rely on millions of synthetic examples in the original reported settings.

**Optimality and efficiency.** We now explore the benefits of optimal neural synthesis in more detail. Specifically, Table 2 compares OPSYNTH with its ablations that perform beam search with varying beam sizes for Test-E. For the purposes of this experiment, we terminate OPSYNTH's search after it has explored a maximum of 2500 states. For the beam-search-based ablations, we terminate search when the beam is filled up with complete programs or the size of partial programs in the beam exceeds a specified threshold.

In Table 2, the column labeled "% Opt" shows the percentage of optimal programs found by the search algorithm. We also show the gap (difference of log probability) between the best programs found by each approach and the optimal programs; this is reported in the column labeled "Gap". Finally, the last three columns show the fraction of solved problems (accuracy), the fraction of programs consistent with the examples, and the number of explored states respectively.

As seen in Table 2, our optimal synthesizer can find the optimal program in 75.5% of cases and solves 46.9% problems after exploring 810 states on average. Beam search with a beam size of 20

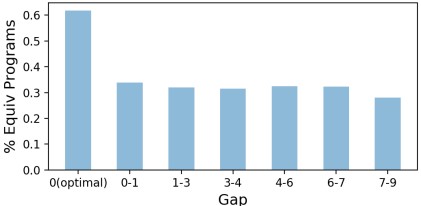

Figure 6: Fraction of programs equivalent to target regex based on score gap with the model-optimal program.

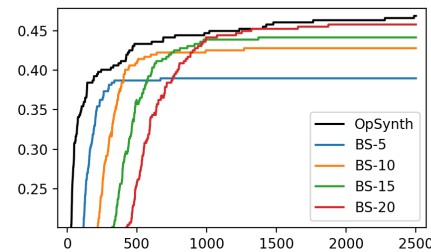

Figure 7: Fraction of solved programs versus the number of explored states.

can only find 66.2% optimal programs, and it solves fewer problems (45.8%) despite exploring $1.8\times$ more states. Although larger beams explore more states than OPSYNTH and find more I/O-consistent programs, they solve fewer problems overall.

We further evaluate the benefit of finding model-optimal programs in Figure 6. Here, we focus only on those programs that are consistent with the input-output examples. The x-axis shows the distance from the optimal program, and the y-axis shows the % of programs that are functionally equivalent to the desired regex. As shown in Figure 6, 62% of optimal programs are equivalent to the target regex, whereas only around 30% of the non-optimal programs match the ground truth functionally.

Finally, Figure 7 plots the fraction of solved problems with respect to the number of states explored. OPSYNTH consistently solves more problems than the other methods given the same budget. Additionally, it also does not require specifying a beam size.

Table 2: Comparison between OPSYNTH and beam-search-based ablations.

|  | %Opt | Gap | %Sol | %Cons | #St |
|---|---|---|---|---|---|
| Beam 5 | 50.4 | 1.11 | 39.0 | 65.1 | 290 |
| Beam 10 | 59.4 | 1.08 | 42.8 | 72.2 | 660 |
| Beam 15 | 63.2 | 0.84 | 44.1 | 76.8 | 1040 |
| Beam 20 | 66.2 | 0.69 | 45.8 | 78.2 | 1430 |
| OpSynth | **75.5** | **0.0** | **46.9** | 75.5 | **810** |

## 6 RELATED WORK

**Natural Language to Logical Forms** Semantic parsing (translating NL to executable logical forms) has been a long-standing research problem in the NLP community (Zelle & Mooney, 1996; Price, 1990). Traditional grammar-based semantic parsers can construct database queries (Zelle & Mooney, 1996; Price, 1990), lambda calculus expressions (Zettlemoyer & Collins, 2005) and programs in other DSLs (Kushman & Barzilay, 2013; Wang et al., 2015). Recent advances in deep learning have explored seq2seq (Jia & Liang, 2016) or seq2tree models (Dong & Lapata, 2016) that directly translate the NL into a logical form, and syntax-based models (Yin & Neubig, 2017) can also inject syntactic constraints. Our approach relies on similar neural modeling to predict the distribution of target programs from NL. However, search is much more complex in our example-guided synthesis setting, whereas prior neural semantic parsers approximate the best solution using beam search (Dong & Lapata, 2016; Yin & Neubig, 2017).

**Optimal Synthesis with Examples** Prior work on PBE considers various notions of optimality using cost functions (Bornholt et al., 2016; Feser et al., 2015; Schkufza et al., 2013) and machine learning (Menon et al., 2013). The first line of work allows users to specify the desired properties of the synthesized program; for instance, smaller program size, lower execution time, or more efficient memory usage. Menon et al. (2013) define optimality as the most likely constructs given a set of examples under a probabilistic context free grammar. In this work, we focus on a new setting where we guarantee the optimality with respect to a neural modal, which can encode specifications such as natural language that are hard to formulate as simple cost functions.

**Multimodal Program Synthesis** There has been recent interest in synthesizing programs using a combination of natural language and examples (Polosukhin & Skidanov, 2018; Chen et al., 2019b; Nye et al., 2019; Andreas et al., 2018; Raza et al., 2015). Specifically, Chen et al. (2020) and Ye et al. (2020b) parse the natural language into an intermediate representation and then use it to guide enumeration, but they do not have any optimality guarantees with respect to the neural model. Kulal et al. (2019) synthesize programs by performing line-by-line translation of pseudocode to code and verify consistency with test cases at the end. However, unlike our approach, their technique enumerates syntactically ill-formed programs, which they address using compiler error localization.

## 7 CONCLUSION

In this paper, we presented a technique for optimal synthesis from multimodal specifications. On a benchmark of complex regex synthesis problems, we showed that this approach is substantially more accurate than past models, and our synthesis algorithm finds the model-optimal program more frequently compared to beam search. While we have evaluated this method in the context of regular expressions, our technique is also applicable for other synthesis tasks.

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

## A  GUARANTEE OF OPTIMALITY

**Theorem 1** (Guarantee of Optimality). *Suppose given a CFG $\mathcal{G} = (V, \Sigma, R, S_0)$, specification $\phi$, natural language $N$ and model $M_\theta$, OPSYNTH returns a program $P^*$. Then, for any program $P \models \phi$, $M_\theta(P) \leq M_\theta(P^*)$.*

*Proof.* Assume $P^*$ is the returned program of OPSYNTH$(\mathcal{G}, \phi, N, M_\theta)$ and there exits a program $P$ such that $P \models \phi$ and $M_\theta(P) > M_\theta(P^*)$. Since $M_\theta(P) > M_\theta(P^*)$, $P$ must have been present in the worklist and considered as a concrete program before the model visited $P^*$. But then, given that $P \models \phi$, then OPSYNTH will return $P$ rather than $P^*$, which contradicts the assumption. $\qquad\square$

## B  CFG FOR REGULAR EXPRESSIONS

We present the CFG for the regex domain language taken from STRUCTUREDREGEX (Ye et al., 2020a). Its correspondence to the constructions in the standard regular expression is shown in the Appendix A of Ye et al. (2020a).

$$
\begin{aligned}
S_0 &\to V_1 \\
V_1 &\to T_1 \mid \texttt{startwith}(V_1) \mid \texttt{endwith}(V_1) \mid \texttt{contain}(V_1) \\
&\quad \texttt{not}(V_1) \mid \texttt{and}(V_1, V_1) \mid \texttt{or}(V_1, V_1) \\
&\quad \texttt{optional}(V_1) \mid \texttt{star}(V_1) \\
&\quad \texttt{concat}^2(V_1, V_1) \mid \texttt{repeat}(V_1, k) \mid \texttt{repatleast}(V_1, k) \mid \texttt{reprange}(V_1, k_1, k_2) \\
T_1 &\to c \mid \texttt{<let>} \mid \texttt{<cap>} \mid \texttt{<low>} \\
&\quad \texttt{<num>} \mid \texttt{<any>} \mid \texttt{<spec>} \mid \texttt{<null>}
\end{aligned}
$$

Figure 8: Regex CFG. Here $k \in \mathbb{Z}^+$ and $c$ is a character class, such as $\texttt{<a>}$, $\texttt{<1>}$, etc.

## C  ENCODING FOR THE INFEASIBLE PROCEDURE FOR REGEX

$$
\Phi^{\{+,-\}}(\texttt{InLang}, y, \mathbf{x}, \mathbf{z}) = (y \wedge (\mathbf{x} \in z_0))
$$

$$
f \in \{\texttt{startwith}, \texttt{endwith}, \texttt{contain}, \texttt{not}, \texttt{optional}, \texttt{star}\} \quad \Phi^{\{+,-\}}(f, y, \mathbf{z}) = (y = f(z_1))
$$

$$
f \in \{\texttt{cat}, \texttt{and}, \texttt{or}, \texttt{repeat}, \texttt{repatleast}\} \quad \Phi^{\{+,-\}}(f, y, \mathbf{z}) = (y = f(z_1, z_2))
$$

$$
f \in \{\texttt{reprange}\} \quad \Phi^{\{+,-\}}(f, y, \mathbf{z}) = (y = f(z_1, z_2, z_3))
$$

Figure 9: $\Phi^{+,-}$ in the regex domain. Here we omit the $T_1$ and $k$ case. The encoding for non-terminal symbols is rule (a) in Figure 5 where $\top = \texttt{star(<any>)}$ and $\bot = \texttt{<null>}$.

We describe our detailed instantiation of the INFEASIBLE procedure described in Section 3.2 in the regex domain. Recall that INFEASIBLE prunes a given partial program $P$ by checking consistency between the approximated semantics and the given examples. In the regex domain, we encode the semantics of a regex in terms of the set of strings it can match. To enable checking consistency between a given example and the regex, given a string $s$, we use the program $\texttt{InLang}(s, P)$ (denoted as $P'$) to represent whether $s$ is in the set of strings that can be matched by $P$. To encode a program $P'$ for consistency checking, we use the set of encoding rules presented in Figure 9 to generate its

---

[2]We note $\texttt{concat}$ as $\texttt{cat}$ in the paper.

over- and under- approximated semantics. In the regex domain, for most of the constructs we can model the precise semantics except for the non-terminal symbols in the partial program.

**Example.** Consider the following partial program $P$: $\texttt{cat(or(<0>}, V_1\texttt{)},\texttt{<1>)}$. To utilize the positive and negative examples for pruning, we first encode the semantics of the program $P'$: $\texttt{InLang}(x, P)$ as follows:

$$(\psi^+, \psi^-) = (\exists \mathbf{z}.y \wedge (x \in z_0 \wedge \psi_0^+[z_0/y])), \exists \mathbf{z}.y \wedge (x \in z_0 \wedge \psi_0^-[z_0/y]))$$

$$(\psi_0^+, \psi_0^-) = (\exists \mathbf{z}.y = \texttt{cat}(z_1, z_2) \wedge \psi_1^+[z_1/y] \wedge \psi_2^+[z_2/y], \exists \mathbf{z}.y = \texttt{cat}(z_1, z_2) \wedge \psi_1^-[z_1/y] \wedge \psi_2^-[z_2/y])$$

$$(\psi_1^+, \psi_1^-) = (\exists \mathbf{z}.y = \texttt{or}(z_3, z_4) \wedge \psi_3^+[z_3, y] \wedge \psi_4^+[z_4/y], \exists \mathbf{z}.y = \texttt{or}(z_3, z_4) \wedge \psi_3^-[z_3, y] \wedge \psi_4^-[z_4/y])$$

$$(\psi_2^+, \psi_2^-) = (y = \texttt{<1>}, y = \texttt{<1>})$$

$$(\psi_3^+, \psi_3^-) = (y = \texttt{<0>}, y = \texttt{<0>})$$

$$(\psi_4^+, \psi_4^-) = (y = \top, y = \bot)$$

We can simplify formulas $\psi^+, \psi^-$ by eliminating the existentially quantified variables:

$$(\psi^+, \psi^-) = (y \wedge (x \in \texttt{cat(or(<0>}, \top\texttt{)},\texttt{<1>)})), y \wedge (x \in \texttt{cat(or(<0>}, \bot\texttt{)},\texttt{<1>)})))$$

Let the positive example be $i = \texttt{"a1"}, o = \texttt{True}$ and let the negative example be $i = \texttt{"01"}, o = \texttt{True}$. According rule (c) of Figure 5, we check if the following formula is unsat:

$$\texttt{True} \wedge (\texttt{"a1"} \in \texttt{cat(or(<0>}, \top\texttt{)},\texttt{<1>)})) \wedge \neg(\texttt{True} \wedge (\texttt{"01"} \in \texttt{cat(or(<0>}, \bot\texttt{)},\texttt{<1>)})))$$

Since the under-approximated semantics of $P$ contains the string $\texttt{"01"}$, this formula is indeed unsat and we are able to prune this partial program.

## D    NEURAL MODEL DETAILS

As described in Section 3.1, our neural model resembles an Abstract Syntax Network (Rabinovich et al., 2017) tailored to fit the regex DSL used in STRUCTUREDREGEX. We show the grammar in Figure 1. As there is no production rule having optional or sequential cardinality, we do not include the "constructor field module" from the ASN in our implementation. We encode the NL using a single layer Bi-LSTM encoder with a hidden state size of 100. In the decoding phase, we set the size of the hidden state in the decoder LSTM as well as the the size of the embedding of $\mathcal{R}(n_j, i_j)$ to be 100. To obtain the contexts, we use the Luong general attention scheme (Luong et al., 2015). To prevent overfitting, we apply a dropout of 0.3 to the all the embedding, outputs of recurrent modules, and context vectors. Our model is trained using Adam (Kingma & Ba, 2015) with a learning rate of 0.003 and a batch size of 25.

## E    SELECTLEAF FUNCTION DETAILS

The SELECTLEAF function selects one non-terminal leaf node in the partial program to expand. We find that when programmatic constraints are integrated into the search process, the order of choose which non-terminal to expand can impact the cost needed to synthesize the target program. We give a concrete example of how the way we select non-terminal leaf nodes to expand can affect the cost of synthesis. Consider a timestep where we obtain the feasible partial program $\texttt{cat}(V_1, V_2)$ from the queue, where both $V_1$ and $V_2$ can be expanded to $\texttt{<0>}$ or $\texttt{<1>}$ with a probabilities 0.9 and 0.1 respectively. Suppose $\texttt{cat(<0>}, V_2\texttt{)}$ is feasible, $\texttt{cat}(V_1, \texttt{<0>)}$ is infeasible, and the only feasible complete program is $\texttt{cat(<1>},\texttt{<1>)}$. If we choose to expand $V_1$ first, then the search procedure goes as follows: $\{(\texttt{cat(<0>}, V_2\texttt{)}, \checkmark) \rightarrow (\texttt{cat(<0>},\texttt{<0>)},\textbf{✗}) \rightarrow (\texttt{cat(<0>},\texttt{<1>)},\textbf{✗}) \rightarrow (\texttt{cat(<1>}, V_2\texttt{)},\textbf{✗}) \rightarrow (\texttt{cat(<1>},\texttt{<0>)},\textbf{✗}) \rightarrow (\texttt{cat(<1>},\texttt{<1>)},\checkmark)\}$, which takes 6 steps. Now, if we expand $V_2$ first, the search procedure is: $\{(\texttt{cat}(V_1, \texttt{<0>)}, \textbf{✗}) \rightarrow (\texttt{cat}(V_1, \texttt{<1>)},\checkmark), \rightarrow (\texttt{cat(<0>},\texttt{<1>)},\textbf{✗}), \rightarrow (\texttt{cat(<1>},\texttt{<1>)},\checkmark)\}$, which only takes 4 steps.

We want to find an order expand the nodes that leads to most effective pruning. We tested the following ways of selecting leaf nodes: (1) pre-order traversal, (2) choosing the highest-level leaf node, (3) choosing the lowest-entropy leaf node. We found that pre-order traversal worked better

than the other strategies in most cases. Given the same budget, using per-order traversal solves more programs while exploring less states compared to the other ways. The superiority of pre-order traversal on the regex synthesis task can be attributed to that our INFEASIBLE function needs concrete terminal leaf nodes to prune effectively, and using pre-order traversal prioritizes deepest nodes and usually yields terminal leaf nodes more quickly than other strategies.

## F    IMPLEMENTATION DETAILS OF THE BASELINES

**ALPHAREGEX**    We implemented the top-down enumerative synthesizer presented in Lee et al. (2016). Although Lee et al. (2016) only uses `<0>` and `<1>` as terminals, here we extended the synthesizer to support most of the ASCII characters.

**DEEPCODER**    We implemented DEEPCODER with a few modifications from its original implementation (Balog et al., 2017). First, we assign each token in the examples with a *class*, and embed the token by both its value and its class. For instance, consider the positive example (`ax4,+`) of the regex `concat(repeat(<low>,2),repatleast(<num>,1)` (2 lower letters followed by 1 or more digits. We assign "a" and "b" with the "`<low>`" class, and assign "4" with the "`<num>`" class. The final embedding of the token "a" is the concatenation of the embedding of the value Emb(a) and the class Emb(`<low>`). We use such combined embeddings for better generalizability. Then, we encode the examples with a Bi-LSTM encoder. Each example is encoded into a hidden vector, which is later max-pooled. Finally, we apply a linear layer on the pooled representation for the whole program, and predict the the the set of probabilities for each of the constructs in the DSL.

We extended ALPHAREGEX to synthesize programs using the probability of constructs obtained from the neural model. In the STRUCTUREDREGEX grammar, we associate each construct with the score returned from the neural network and calculate the score of a partial program by summing up the score of all the constructs that are used in the partial program. We specify the synthesizer to prioritize exploring the partial programs with the highest score so far.

Recall that in Section 5 that DEEPCODER doesn't achieve high performance in the STRUCTUREREGEX dataset. Since most of the constructs are recursive in the regex language and DEEPCODER search is essentially doing a depth-first search, the synthesizer first needs to exhaustively check all possible programs associated with the highest probability constructs before it can move on to explore those programs with any other constructs. For example, suppose the `concat` has the highest probability and the synthesizer explores programs up to maximum depth 5, the synthesizer will prioritize exploring programs like `concat(concat(concat(concat(<low>))))` and searching in this way does not help the synthesizer to find the ground truth regex.

**ROBUSTFILL**    We implemented the ATTENTION A model from ROBUSTFILL (Devlin et al., 2017), which predicts programs given I/O examples. We encode the I/O with the the same I/O embedding and I/O encoder used in our implementation of DEEPCODER. We replaced the LSTM decoder in the original implementation with our ASN decoder. During decoding, we extract a context vector from each of the examples provided in the example set, and pool them with max-pooling as the final context vector. The probability distribution over rules for node $n$ is then given as:

$$\text{Attn}(h_n, \text{context}(\phi) = \text{MaxPool}(\{\text{Attn}(h_n, \text{context}(e))\}_{e \in \mathcal{E}})$$
$$p_\theta(r|n, P, N) = \text{softmax}(\text{FFNN}(h_n; \text{Attn}(h_n, \text{context}(\phi))))$$

We set the size of value embedding and class embedding to be 50, and the size of hidden state in encoder Bi-LSTM and LSTM in ASN to be 100.

**TREESEARCH**    As the code of TREESEARCH (Polosukhin & Skidanov, 2018) is not publicly available code, we implemented our own version of TREESEARCH on top of TRANX which is reported to be more powerful than the originally used SEQ2TREE on various datasets (Yin & Neubig, 2018). During search, we set the threshold to be $10^{-5}$, and the max queue size to be 100.

**OPSYNTH$^{+\mathcal{R}}$**    We naturally combine OPSYNTH and ROBUSTFILL by concatenating the context vectors from NL and examples, as in Section 4. The hyper-parameters for the NL encoder are the same as those for the base synthesizer, and the hyper-parameters for the I/O encoder are the same as ROBUSTFILL.

