# OpenReview forum: "Optimal Neural Program Synthesis from Multimodal Specifications"
_ICLR.cc/2021/Conference — Reject_

### Official Review · AnonReviewer2 · 2020-10-26
**Official Blind Review #2**

**Rating:** 6
**Confidence:** 5

**Review:**

Summary
-------------
A method for synthesizing programs from a combination of natural language specifications and formal specificatios (or, more concretely, input/output examples) is presented. It combines a top-down, grammar-based expansion strategy with a deduction component that can help to rule out partial programs that cannot be completed into a correct program. Experiments show small but noticeable improvements over baselines on a dataset of regular expression synthesis benchmarks.

Strong/Weak Points
-------------
* (+) Paper attempts to integrate neural and symbolic methods more deeply than just using one as filter for the other.
* (+) Experiments with relevant baselines seem clean and answer the most important questions.
* (-) Lack of comparison with highly related work that combines neural generation approaches with deductive methods to simplify the search space. While most of these methods have not been developed for the multi-modal setting. Handling multi-modal specifications would be similarly straightforward as the extension from OpSynth to OpSynth+R described in Appendix E. Examples of relevant work:
   * "Neural-guided deductive search for real-time program synthesis from examples" (Kalyan et al.) uses version space algebras to determine if a partial program can be completed to a correct program and a neural network conditioned on the partial program to select the next most-likely expansion step.
   * "Program Synthesis using Conflict-Driven Learning" (Feng et al) uses a CDCL-like procedure to rule out infeasible parts of the search space with a ML-based heuristic for chosing next expansion steps.
   * Execution-guided approaches ("Execution-guided neural program decoding" (Wang et al.), "Execution-guided neural program synthesis" (Chen et al.) which check if the partial program generated so far is compatible with the examples. In the regex space, this is naturally applicable to left-to-right generation approaches by checking if the partial regex generated so far matches a prefix of the input example.
* (-) For the ICLR audience, Sect. 3.2 may be at a too formal level (and especially, familiarity with the inference rule notation in Fig. 5 should not be assumed). I have doubts that this _helps_ readers to understand and due to the space limitations, it doesn't make things exact either (e.g., it took me a while to understand that $\Phi^+$/$\Phi^-$ implicitly consume $\mathbf{z}$; and the "definitions" for $\Phi^{+,-}$ in Appendix B/Fig. 9 are not precise as they just map to some undefined underlying regular expression language)

Recommendation
-------------
I think this is a borderline paper. Overall, it is in a crowded space with many competing approaches and does not do a great job at differentiating itself (see above), which makes it hard to judge its contribution clearly. The experimental results are an indication that the method works, but some questions (e.g., does using ASN instead of a stronger neural generator model make things worse) remain.

Questions
-------------
* [See above re related work that could be compared to]
* Appendix D states that implementing SelectLeaf using a pre-order traversal works best, which somewhat negates the argument for an approach that is invariant to derivation order (in Sect 3 / 3.1). If pre-order works best anyway, building on top of TranX (or "Generative Code Modeling with Graphs" (Brockschmit et al)) would be possible as well, which have shown better results than ASN in generative code modelling. Did you experiment with such models?

Detail Feedback
-------------
* Page 3: "Wwe" -> "We"
* Fig. 9:  second/third equation are lacking a closing ")"; "or" appears twice in the third equation.

---

> ### Author Response · Authors · 2020-11-14
> **Response to AnonReviewer2**
>
> Thanks for the detailed feedback. We address the main points of concern below.
>
> *Q: Lack of comparison with highly related work that combines neural generation approaches with deductive methods to simplify the search space. While most of these methods have not been developed for the multi-modal setting. Handling multi-modal specifications would be similarly*
>
> A: Extending such approaches to work in our setting requires substantial effort and is outside the scope of the paper. As a comparison to these approaches:
>
> The scoring of a production rule in Kalyan et al. (2018) only depends on the specification but not on the partial program enumerated so far. Also, the search procedure in this paper has no optimality guarantee, which is the reason they propose two types of controller to handle the performance and  generalizability tradeoff.
>
> Feng et al. (2018) rely on using an SMT solver, which can be computationally costly and is not always easy to apply to a domain such as regex. This work also doesn’t guarantee optimality in its search procedure with respect to the neural model.
>
> Our ASN+P model is basically the same as execution-guided approaches (Wang et al, 2019), where we prune out infeasible partial regexes in the decoding procedure using the proposed pruning technique that is more powerful than prefix matching.
>
> *Q: For the ICLR audience, Sect. 3.2 may be at a too formal level (and especially, familiarity with the inference rule notation in Fig. 5 should not be assumed).*
>
> A: We can include a worked example in appendices. We tried to strike balance between rigorously defining our procedure and making the paper approachable given the space limit.
>
> *Q: If pre-order works best anyway, building on top of TranX (or "Generative Code Modeling with Graphs" (Brockschmit et al)) would be possible as well, which have shown better results than ASN in generative code modelling. Did you experiment with such models?*
>
> A: Yes. The TreeSearch model performs tree search using a neural model built on top of TranX, which is not as good as OpSyth-P.
>
> Conceptually, we believe this happens for the following reason (repeating our answer to reviewer #4): Our work uses a functional DSL, whereas Brockschmidt et al. generate C# programs. Functional programs are more likely to be built out of semi-independent pieces, whereas the expression completion task from Brockschmidt et al. has stronger dependence between sibling nodes when variables are reused across expressions in the more procedural style. We believe this difference is important when considering what kind of factorization in the model is likely to be most effective.
>
> We also note that models like TranX are not universally better than ASN: on the HearthStone dataset, ASN reports a significant advantage over the action-sequence based model (Yin & Neubig, 2017).

---

> > ### Comment · AnonReviewer2 · 2020-11-23
> > **Respone^2 to AnonReviewer2**
> >
> > Thank you for your reply and the paper update. Let me note that you seem to have missed my comments on the typos in Fig. 9, which was not updated.
> >
> > > Our ASN+P model is basically the same as execution-guided approaches (Wang et al, 2019),
> > > where we prune out infeasible partial regexes in the decoding procedure using the proposed
> > > pruning technique that is more powerful than prefix matching.
> >
> > Now my interest is piqued: I tried to came up with a simple example and a prefix that could be pruned by your proposed method, but not by the prefix-matching trick. The latter is a bit tricky in the presence of "or(...)" but doable, and so the only examples I can come up with are either using non-standard regular expression primitives (contain/endwith/repeat/...) or fail because the functional representation of "star(R)" is not amenable to the prefix trick (but writing it as "R*" would work well).
> >
> > So maybe it would be helpful for the reader to illustrate the difference by an example, to make it easier to understand the benefit of your proposed method?
> >
> > > > For the ICLR audience, Sect. 3.2 may be at a too formal level (and especially, familiarity
> > > > with the inference rule notation in Fig. 5 should not be assumed).
> > > We can include a worked example in appendices. We tried to strike balance between
> > > rigorously defining our procedure and making the paper approachable given the space limit.
> >
> > Using a formalism that the audience is not familiar with is not a "rigorous definition", it just becomes visual noise that people ignore. This is not POPL/PLDI and hence I'd strongly recommend adapt your presentation to the venue and move the inference rules into the appendix and use the additional space for an example instead.

---

> > > ### Author Response · Authors · 2020-11-23
> > > **Response to AnonReviewer2**
> > >
> > > Thanks for the detailed reply.
> > >
> > > > I tried to came up with a simple example and a prefix that could be pruned by your proposed method, but not by the prefix-matching trick.
> > >
> > > Our approach can prune those partial regexes that do not have deterministic prefixes, e.g., contain(concat(<let>,?)), and(endwith(?),?), whereas simple prefix-matching is not effective in these cases.
> > >
> > > > So maybe it would be helpful for the reader to illustrate the difference by an example, to make it easier to understand the benefit of your proposed method?
> > >
> > > > I'd strongly recommend adapt your presentation to the venue and move the inference rules into the appendix and use the additional space for an example instead
> > >
> > > Thanks for the feedback about the presentation. While we will not attempt to rush a major update to the presentation of the paper during the review period, we will explore how to rewrite this in any future version.

---

### Official Review · AnonReviewer1 · 2020-10-28
**Interesting results on a specific task. Possible writing improvements**

**Rating:** 5
**Confidence:** 4

**Review:**

The paper shows an interesting and novel combination of neural and constraint-based synthesis that despite using a relatively simple neural model, manages to improve over more involved neural synthesis techniques. This is a significant point in the space of such algorithms and an important research result.

Several important systems were compared, including non-neural results. One interesting question is in what cases Sketch found a solution compatible with the input/output examples, but it was not the desired solution for the task.

There are a few areas for improvement of the paper, but overall it is well written and shows interesting results:

- The technique is described very generally, but there was only one dataset and one task for which it improved the results over the baselines. This task has the advantage that it contains an elaborate “Infeasible” procedure that is usually not present in general DSLs, but it does not seem that this is the main reason why the technique works well. Judging from the ablated results, it seems that the main reason seems to be that the other neural baselines are baselines are a poor fit for this specific task.

- While the approach talks about speed, it only includes the number of steps in its results. Was there any improvement in terms of wall-time say in comparison to Sketch? You seem to be giving 90 seconds to it. How long does it take for OpSynth to run?

- Some improvements in the definitions and the writing are possible:

page 2: In f( s_0 .. s_n) where as later it talks s_1 to s_n
page 3: Wwe

Definition 3.1 seems inaccurate and does not imply that a path must start at the root. If it does not, then the function pi cannot be defined. Also n_k is not the same as n_k in the examples.
Definition 3.2 defines assigning rules to nodes, but this is not part of a tree or a grammar. It looks like it tries to make applying a production rule in two steps, but it is not clear why this is needed.
Supp is not defined in the algorithm.

---

> ### Author Response · Authors · 2020-11-14
> **Response to AnonReviewer1**
>
> We would like to thank the reviewer for constructive comments and clarify the following concern:
>
> *Q: One interesting question is in what cases Sketch found a solution compatible with the input/output examples, but it was not the desired solution for the task.*
>
> A: In some cases, Sketch finds some programs that are very similar but not equivalent to the target program. For instance, Sketch may fail to include some subpart in the regex surrounded by optional (e.g., optional(<let>)) when the examples do not cover enough corner cases.
>
> *Q: There was only one dataset and one task for which it improved the results over the baselines.*
>
> A: To the best of our knowledge, the StructuredRegex dataset is the only existing multimodal program synthesis dataset that contains natural language written by real humans. The other relevant dataset, AlgoLisp, only contains synthetic natural language, which is the reason that existing systems can already achieve near-perfect performance on this task.
>
> *Q:  Was there any improvement in terms of wall-time say in comparison to Sketch?*
>
> A: We’ve added wall clock time in our revision. It takes around 20-25s for OpSynth to search for 2500 states, which improves the efficiency by 3X on wall-clock time compared to Sketch.
>
> *Q: Definition 3.1 seems inaccurate and does not imply that a path must start at the root.*
>
> A: All paths must start at the root -- we will clarify this.
>
> *Q: Why does Definition 3.2 define assigning rules to nodes, which is not part of a tree or a grammar. It looks like it tries to make applying a production rule in two steps.*
>
> A: We always start from a root node and choose the production rule to apply on it. Everytime a production rule is applied, it creates new undetermined nodes (its child fields). So the new nodes are instantiated as a rule is applied, not as a separate step in the process.

---

### Official Review · AnonReviewer4 · 2020-10-30
**Limited technical novelty but a strong end-to-end contribution with provable optimality**

**Rating:** 7
**Confidence:** 4

**Review:**

The work presents a technique for synthesizing program from a combination of NL and input-output examples. It is guaranteed to find the best-possible program under a trained NL->AST model that satisfies the given examples. The technique assumes a particular kind of NL->AST model from the literature and a pruning mechanism for infeasible partial programs, integrated into best-first search. On a recent dataset of multi-modal regular expression problems, the technique significantly outperforms both purely-neural, purely-symbolic, and prior multi-modal baselines, although it only slightly outperforms its own ablations.

## Strengths & Weaknesses

The main contribution is using best-first search to guarantee optimality wrt the semantic parsing model, which, in turn, required fixing the model class to ASN. The infeasibility pruning seems to repeat the procedure of Chen et al. (PLDI 2020). Nevertheless, the end-to-end formulation is a valuable contribution to the program synthesis community even if its technical novelty is limited.

As far as I'm aware, this is the first multi-modal program synthesis method that has an optimality guarantee under a neural model. Other approaches to optimal program synthesis exist, but they focus on manual cost functions. In theory, best-first search could be integrated into other methods, but this is the first work that presents such a method end to end.

The paper is written clearly and with plethora of examples. One exception are the inference rules for pruning in Figure 5, which introduce a significant abstraction that does not become clearer until Appendix B. However, space constraints are understandable and unavoidable. In absence of space, I would suggest swapping the two and presenting a concrete example of infeasibility (plus perhaps pseudocode) in Section 3 and domain-independent formalism in the Appendix.
Given that the paper only evaluates itself on a single application domain, a domain-independent general abstraction of pruning without a regex-related example in the main paper body seems excessive.

The method clearly outperforms both single-modality and prior multi-modal baselines. The ablation experiments are also quite insightful. Interestingly, best-first search only adds 0.9% of accuracy on Test-E over beam search. The main gains in the approach seem to come from the choice of ASN as a semantic parsing model and the feasibility pruning, not from the best-first formulation.

## Questions/Suggestions

- As stated in the Appendix D, the authors expand the nodes using the pre-order traversal in practice. This is a fixed order of expansion, the same one as TranX. The models, of course, differ - ASN conditions its prediction on the AST path whereas TranX conditions on the expansion path and the parent node. In Table 1, ASN+P outperforms TranX+P by 7.7 points. However, prior works (Yin & Neubig, 2019; Brockschmidt et al., 2019) report an opposite ordering, with TranX-style models outperforming ASN-style models on many different datasets. Do you have any intuition why the performance of these two baselines so drastically flips on your task?

- Could you formally state and prove the statement of optimality wrt the semantic parsing model as a theorem? It is intuitively true, but the paper (otherwise impressively rigorous) currently only posits optimality at a very high level in a single sentence on page 5.

---

> ### Author Response · Authors · 2020-11-13
> **Response to AnonReviewer4**
>
> Thanks for the valuable comments and suggestions!
>
> *Q: The main gains in the approach seem to come from the choice of ASN as a semantic parsing model and the feasibility pruning, not from the best-first formulation.*
>
> A: Most importantly, the best-first formulation enables a large improvement in the search speed, cutting the search cost by roughly 2X. It does also improve the accuracy by 1.8%.
>
> *Q: Prior works (Yin & Neubig, 2019; Brockschmidt et al., 2019) report an opposite ordering [between ASN and TranX]. Do you have any intuition why the performance of these two baselines so drastically flips on your task?*
>
> A: Our work uses a functional DSL, whereas Brockschmidt et al. generate C# programs. Functional programs are more likely to be built out of semi-independent pieces, whereas the expression completion task from Brockschmidt et al. has stronger dependence between sibling nodes when variables are reused across expressions in the more procedural style. We believe this difference is important when considering what kind of factorization in the model is likely to be most effective.
>
> We also note that models like TranX are not universally better than ASN: on the Hearthstone dataset, ASN reports a significant advantage over the action-sequence based model (Yin & Neubig, 2017).
>
> *Q: Could you formally state and prove the statement of optimality wrt the semantic parsing model as a theorem?*
>
> A: We now provide the proof in the appendix.

---

### Official Review · AnonReviewer3 · 2020-11-03
**Promising approach, but contributions unclear, limitations under explored, and experiments don’t demonstrate what they should.**

**Rating:** 4
**Confidence:** 4

**Review:**

## Summary
This paper proposes an approach to program synthesis that aims to incorporate information from different modalities, focusing on combining input-output examples with natural language specifications. They formulate it as a form of constrained optimal program synthesis.  The constraints are in the form of positive and negative examples, which the program must satisfy.  The optimality criteria is in the form of maximizing the conditional probability of a program $P$ given a natural language description, as specified by a conditional distribution that is learned from data.

## Overview

Overall:
- The use of program analysis methods within neural program synthesis is promising.
- I found no technical flaws with this paper.
- The paper needlessly combines different concepts, and doesn’t clearly disambiguate what is it’s contribution.   The main contribution of this paper is in pruning partial programs, but that is obscured under the framing of multimodal learning.
- Both the benefits and limitations of an abstract interpretation approach are under explored.

It is difficult to tell exactly what the claims of the paper are, as they are not clearly specified.  Nevertheless, there are three different concepts in this paper that are worth separating:

- The representation of the distribution over expressions that supports enumerating programs in order of decreasing probability.  Consequently the first program in this enumeration that satisfies the positive and negative examples is optimal.
-  The use of abstract interpretation to prune infesabile partial programs.  This means that conditioning $p_\theta$ on the fact that $P$ must satisfy positive and negative examples can be done more efficiently than simply rejecting complete programs.
- The formulation of optimal multimodal program synthesis in these terms.

(please correct me if I am missing something or mischaracterizing the contributions)

The representation used (Abstract Syntax Networks) is not a contribution of this work.  It is not entirely clear to me whether the authors are claiming the enumeration strategy that produces the most probable program first as a contribution.  If so, there is existing work both in conventional program synthesis (e.g. [1, 2]) and neural program synthesis (e.g.[3]) that enumerates programs in order of probability.  Does this approach present anything new over these?

The formulation is fine.

Given all of this, it seems that the primary contribution of this paper is the use of a form of abstract interpretation to prune partial programs.  There is much merit to this idea and approach.  In particular, there are potentially very large gains to be had in efficiency.  I have a number of concerns though:

**Writing:** Given the importance of this section, the abstract interpretation is very poorly explained.  Terms in the derivation tree (e.g. Root, SubTree, Children) are not specified.  Several notations (such as expression substitution) which are likely unfamiliar to a machine learning audience are not explained.

The authors have also not cited any of the work that uses abstract interpretation for program synthesis (e.g. [4,5,6])

**Imprecision:** one of the major problems in abstract interpretation is imprecision — the over approximation can include too many in feasible solutions, and the under approximation can exclude too many.  Familiarity with abstract interpretation would lead most to have a healthy dose of skepticism that the approach can be used naively for program synthesis, without vary careful selection of the abstract domains.  This is because for most abstract domains in most non-trivial examples, imprecision would lead to an inability to prune virtually any programs.

Carefully choosing abstractions for specific domains can be done in some situations, which is precisely what the regex example has done.  They present an abstract interpretation framework that is parameterized using the $\psi$ function, but the only example is in terms of regular expressions which permit precise abstractions.

This is not to say that focusing on restricted applications isn’t worthwhile, but the authors should take a lot more care to both explore and explain the limitations of abstract interpretation and the difficulties that one would face in actually applying their framework to other DSLs.

**Results** The results are a little hard to interpret.  $\text{OpSynth}$ solves 5% more (of the total number of) solutions than the ablated $\text{OpSynth}^{-\mathcal{P}}$.  However, this ablated version has nothing checking for consistency with the actual data.  In addition to this, the authors should present in Table 1 data for OpSynth with a routine that does check for consistency, such as the beam search approaches.

When comparing against different approaches, it is impossible to tell from the data if failures are due to timing out or due to finding a program that doesn’t generalize to the test data.  If it is the former, then the results may look very different with different timeout thresholds.

Wall clock time is missing from evaluation, which is as important since neurosymbolic methods can be orders of magnitude slower in real time than conventional methods.

Taking a step back, the primary problem is that the bulk of the work done is in the ASN network, and importance of the pruning cannot be discerned from this.  The advantages in Table 2 of OpSynth may be very much diminished with a greater beam size / threshold.

This is all to say that I cannot tell from this data that OpSynth has real advantages over ASN combined with simple enumeration.

## Questions

- OpSynth finds 75.5% of the optimal programs.  Is this optimality with respect to the trained model $p_\theta$?  If so, according to your approach, the remaining 25.5% of failures must be due to timeouts.  Is this correct?
- What exactly is the “fraction of solved problems”?  How could it be less than the percentage of optimal programs found?  Is this performance on a test set?

## Typo
Wwe -> We

## References
[1] Lee, Woosuk, et al. "Accelerating search-based program synthesis using learned probabilistic models." ACM SIGPLAN Notices 53.4 (2018): 436-449.
[2] John K Feser, Swarat Chaudhuri, and Isil Dillig. Synthesizing data structure transformations from input-output examples. In PLDI, 201
[3] Ellis, Kevin, et al. "Dreamcoder: Growing generalizable, interpretable knowledge with wake-sleep bayesian program learning." arXiv preprint arXiv:2006.08381 (2020).
[4] Rishabh Singh, Armando Solar-Lezama, Synthesizing data structure manipulations from storyboards, 2011(bibtex)
[5] Martin T. Vechev, Eran Yahav, Greta Yorsh, Abstraction-guided synthesis of synchronization
[6] Wang, Xinyu, Isil Dillig, and Rishabh Singh. "Program synthesis using abstraction refinement." Proceedings of the ACM on Programming Languages 2.POPL (2017): 1-30.

---

> ### Author Response · Authors · 2020-11-13
> **Response to AnonReviewer3**
>
> Thank you very much for your detailed review and suggestions! We have included missing results and we would like to explain the following points:
>
> **[Contributions]**
>
> *Q: What are the main contributions of the paper?*
>
> A: The main contribution is an **optimal** synthesis technique for **multi-modal** program synthesis (combining examples and natural language). While the ideas of optimality and multi-modality have been independently explored in prior work, this is the first paper that guarantees optimality with respect to an NLP model in the multi-modal context. The prior work that you list for optimal synthesis is all in the pure PBE setting. Multimodality imposes the additional challenge of needing a complex neural model to handle open-ended natural language inputs; standard autoregressive seq2seq models do not typically admit efficient inference. Our techniques are novel in that we show how to do efficient inference in this setting.
>
> **[Abstract Interpretation]**
>
> *Q: Discuss the difficulties of applying abstract interpretation to other domains.*
>
> A: We agree with the reviewer that the choice of abstract domains is crucial: if the abstract domain is too imprecise, it may not have good pruning power; on the other hand, if it is too precise, its overhead may outweigh the benefits of pruning. This is why our algorithm is parametrized by over- and under-approximating abstract semantics. For the regex domain, we provide a concrete instantiation of these semantics and show that it is helpful in practice. Prior work in the context of pure PBE has proposed useful abstract domains in other settings, such as table transformations [1] and tensor and string manipulations [2]. Furthermore, prior work also proposes automated techniques for learning useful abstract semantics for a given DSL and synthesis engine [3]. Thus, choosing a suitable abstract domain for a specific DSL is orthogonal to the contributions of this work.
>
> [1] Feng, Yu, et al. “Component-based synthesis of table consolidation and transformation tasks from examples”
> [2] Wang, Xinyu, et al. “Program synthesis using abstraction refinement”
> [3] Wnag, Xinyu, et al. “Learning Abstractions for Program Synthesis”
>
> **[Results]**
>
> *Q: Table 1 should present data for OpSynth-P with a routine that does check for consistency, such as the beam search approaches.*
>
> A: Such results are already there.  In particular, Seq2Seq+P and TranX+P perform model-guided beam search with pruning in exactly the same way as done in OpSynth. As we show in Table 1, augmenting beam search with pruning increases the number of benchmarks solved by 7 to 11 percent absolute.  In addition, OpSynth-P does check for consistency with examples, but only of concrete programs.
>
> *Q: When comparing against different baselines, it’s impossible to tell whether failures are due to timeouts or failure to generalize?*
>
> A: We have included such results in Table 2 for OpSynth and the ablation ASN+P. OpSynth finds 75.5% of the optimal programs with respect to the neural model. The remaining 24.5% of the errors are due to timeouts. 28.6% of the errors are due to finding spurious programs (consistent with examples but not the user-specified program).
> By contrast, ASN+P finds roughly the same number of consistent programs if allowed to explore many more states, but the programs are not optimal and thus solve fewer problems.  We have also provided the updated results for the other baselines in the revision.
>
> *Q: Wall clock time evaluation is missing.*
>
> For all the neural models in the last two sections of Table 1, the wall clock time follows the number of states very closely, because the neural computation and pruning costs are similar. We include the average time used in our revision. The baselines are the same order of magnitude as these methods in terms of wall clock, and are much less effective.
>
>
> *Q: The advantages in Table 2 of OpSynth may be very much diminished with a greater beam size/threshold.*
>
> A: Table 2 shows that the optimal synthesis approach does a better job of finding the optimal program than beam search even when beam search explores more states. Of course, as the beam size grows larger and larger, the results will get closer, but beam search will take much longer.
>
> *Q: What exactly is the “fraction of solved problems”? How could it be less than the percentage of optimal programs found? Is this performance on a test set?*
>
> A: In the context of multimodal program synthesis, a program that is consistent with examples may not be the desired program. (In contrast, most papers in the  PBE setting consider a solution to be valid as long as it satisfies all the examples.) Thus, the fraction of solved problems reports the percentage of cases where we find the ground truth program. By contrast, an optimal program is optimal with respect to the model, which may not be correct.

---

### Author Response · Authors · 2020-11-14
**Updates in the Result Section**

We would like to thank all the reviewers for the valuable comments and suggestions. We’ve updated our paper based on the feedback, particularly the results section. Specifically, we now include the fraction of problems for which we can find an I/O-consistent program and the average wall-clock time used for each of the approaches. We believe the newly added statistics better distinguish the reasons for failures and demonstrate the advantages of our approach, particularly in terms of efficiency.

---

### Decision · Program_Chairs · 2021-01-07
**Final Decision**

**Decision:**

Reject

**Comment:**

The paper proposes a new multimodal neuro-symbolic technique for synthesizing programs. The specification is given in natural language (soft constraints) and input-output examples (hard constraints). The multimodal program synthesis is formulated as a constrained maximization problem where the goal is to find a program maximizing the conditional probability w.r.t. the natural language specification while satisfying the input-output examples. The proposed technique is evaluated on a multimodal synthesis dataset of regular expressions, and significant performance gains are shown w.r.t. the state-of-the-art synthesis methods. Overall this is an important direction of research, and the paper presents significant results in the space of multimodel program synthesis.

I want to thank the authors for actively engaging with the reviewers during the discussion phase. The reviewers generally appreciated the paper's ideas; however, there was quite a bit of spread in the reviewers' assessment of the paper (scores: 4, 5, 6, 7). In summary, this is a borderline paper, and unfortunately, the final decision is a rejection. The reviewers have provided detailed and constructive feedback for improving the paper. In particular, the authors should more clearly describe the paper's primary contributions, compare their technique with related work that combines neural generation approaches with deductive methods, and simplify the presentation of technical sections. This is exciting and potentially impactful work, and we encourage the authors to incorporate the reviewers' feedback when preparing future revisions of the paper.